# Breaking the Dormancy of Snake’s Head Fritillary (*Fritillaria meleagris* L.) In Vitro Bulbs—Part 2: Effect of GA_3_ Soaking and Chilling on Sugar Status in Sprouted Bulbs

**DOI:** 10.3390/plants9111573

**Published:** 2020-11-13

**Authors:** Marija Marković, Milana Trifunović Momčilov, Branka Uzelac, Olga Radulović, Snežana Milošević, Slađana Jevremović, Angelina Subotić

**Affiliations:** Department of Plant Physiology, National Institute of Republic of Serbia “Siniša Stanković”, University of Belgrade, Bulevar despota Stefana 142, 11060 Belgrade, Serbia; milanag@ibiss.bg.ac.rs (M.T.M.); branka@ibiss.bg.ac.rs (B.U.); olga.radulovic@ibiss.bg.ac.rs (O.R.); snezana@ibiss.bg.ac.rs (S.M.); sladja@ibiss.bg.ac.rs (S.J.); heroina@ibiss.bg.ac.rs (A.S.)

**Keywords:** dormancy changes, *F. meleagris*, soaking bulbs

## Abstract

The bulb is the main propagation organ of snake’s head fritillary (*Fritillaria meleagris* L.), a horticulturally attractive and rare geophyte plant species. In this study, we investigated the effect of soaking bulbs in GA_3_ solution (1, 2, and 3 mg L^−1^) combined with low-temperature treatment (7 °C) on breaking the dormancy of in vitro bulbs. Sugar status (total soluble sugars, glucose, and fructose content) was analyzed in different parts of the sprouted bulbs. The results showed that the soluble sugar concentration was highest in bulbs soaked in GA_3_. The main sugar in fritillary bulbs was glucose, while fructose content was much lower. Glucose concentration dramatically increased after bulb chilling (7 °C), and its accumulation was predominantly detected in the lower sprout portion during the first weeks of sprouting. Sugar concentration was significantly lower in nonchilled bulbs, which indicates the importance of low temperature in bulb development and sprouting.

## 1. Introduction

Bulbs are reproductive organs used for vegetative propagation in geophytes, such as fritillary. Geophytes are characterized by a very short active aboveground growth period in the spring, whereas through the winter period or dry season of the year, they survive in the form of an underground storage organ, e.g., bulb in fritillary. Fritillary bulbs contain many bioactive compounds with medicinal properties, and the whole plant is very attractive from the perspective of horticulture [1]. The physiology of geophytes’ dormancy has not been fully elucidated. Dormancy implies a quiescence period without any visible morphogenesis in vivo as well as in vitro.

Carbohydrates play a very important role in bulb dormancy and development. They also participate in many physiological processes and biological syntheses [2]. Soluble sugars are the prime source of energy, which can be transported from and used in the different plant parts from the primary place of accumulation—storage organ. The distribution and translocation of sugars were documented earlier in potato, *Tulipa*, and lily [3,4,5], but a similar study in fritillary bulbs has never been documented in the literature.

Hormonal balance was already proved to be a crucial factor for bulb formation, dormancy, and sprouting [6], but the principle and physiology of hormonal pathways are unknown in fritillary. With the aim to upgrade the production, regeneration ability, dormancy release, and bulb sprouting in the horticulture and crop industry, phytohormonal balance needs to be explored. Endogenous gibberellins were detected at high levels in sprouting bulbs and their increased concentration inhibited the beginning of dormancy [7]. In the majority of geophytes, low levels of endogenous gibberellins induce dormancy [8]. Therefore, the logical way to investigate bulb dormancy was gibberellin treatment in combination with various temperature treatments. Variation in the concentration of exogenously applied plant growth regulators (PGRs), their composition, and their method of application were the most effective experimental ways to achieve this goal [5,8]. There are some results concerning bulb physiology when gibberellins were applied in the nutrition medium, but its application in the form of soaking solution remained mostly unexplored. In this study, we focused our research on sugar content in sprouted buds after soaking bulbs in gibberellic acid (GA_3_) solution.

Breaking the bulb dormancy and sprouting can be increased by a short treatment with gibberellins with or without chilling treatment. This effect of gibberellin was noticed in lily [9] and fritillary bulbs, which were soaked in GA_3_ solution [1]. The authors detected an increase in the percentage of sprouted bulbs when they were treated with gibberellins even if bulbs were not exposed to low temperatures. *F. meleagris* bulbs start to sprout but do not grow further after short gibberellin treatment without low temperature. This mechanism was not studied in detail but it can be assumed that the gibberellins caused certain biochemical and morphological changes in bulbs. Formation of storage organs such as corms, tubers, or bulbs begins with the swelling of various organs, bud, leaf, shoot, or root [8]. Consequent changes are visible at the macro and micro level and are strongly linked with carbohydrate accumulation, dormancy, and sprouting. The development of new bulbs and their sprouting after cold treatment is a complex many-step process that includes cell division, elongation, and expansion [5], ultimately resulting in sprout elongation, swelling, and the accumulation of nutritive compounds (sugars).

The aim of this study was to investigate the effect of bulb soaking in GA_3_ solution and chilling on sugar distribution in different parts of bulbs during breaking the dormancy.

## 2. Results

### 2.1. Effect of GA_3_ Soaking on Bulb Development and Sugar Content

Control bulbs, soaked for 24 h in sterile water, continued further development and multiplication on plant growth regulator (PGR)-free MS (Murashige and Skoog) medium. After four weeks of culture, several new bulblets were developed in control (Figure 1A). The regenerated bulblets were light green, while soaking *F. meleagris* bulbs in GA_3_ solution (1, 2, and 3 mg L^−1^) for 24 h resulted in minor changes in bulb morphology. After two weeks of culture, small bulbs or bulb scales also appeared on the outside of the main bulb soaked in GA_3_ solution (Figure 1B,C). These newly formed bulbs were greener and longer than the corresponding control. After four weeks of culture, in bulbs soaked in GA_3_, bulb sprouting was achieved at all applied concentrations (Figure 1D–F). Bulbs soaked in 3 mg L^−1^ GA_3_ solution formed more compact, greener bulbs (Figure 1F) compared to control or lower GA_3_ concentrations (Figure 1D,E). A higher level of GA_3_ (2 and 3 mg L^−1^) in the initial soaking solution resulted in necrotic changes at the edge of the scales.

Soluble sugar (Figure 2A), fructose (Figure 2B), and glucose (Figure 2C) contents were analyzed in bulbs and the lower and upper sprout portion four weeks after soaking in GA_3_ solution (1, 2 and 3 mg L^−1^). Results were compared with control bulbs soaked in water for 24 h (Figure 2A–C). We found significant changes in the content of all sugars, in different parts of sprouted bulbs, affected by one day of soaking in GA_3_.

The content of soluble sugars in bulbs was significantly increased in bulbs soaked in GA_3_ (Figure 2A) in comparison to control, water-soaked bulbs. A similar increase in total sugars was observed also in lower sprout portions after GA_3_ treatment. The sugar content in lower sprout portions soaked in 1 mg L^−1^ GA_3_ was almost two-fold higher than in the control. Contrary to this, total sugar content in the upper sprout portions was significantly lower in GA_3_-soaked bulbs, compared to control.

The highest fructose content was observed in control bulbs (114.22 mg/g). Fructose content gradually decreased in bulbs soaked in GA_3_ (Figure 2B). In lower sprout portions, a change in fructose content was observed only after treatment with higher concentrations of GA_3_ (2, 3 mg L^−1^). As well as the total sugar content, the fructose content gradually decreased in upper sprout portions of GA_3_-treated bulbs. The lowest fructose content was observed in bulbs soaked in 3 mg L^−1^ GA_3_.

GA_3_ soaking affected glucose content in snake’s head fritillary bulbs (Figure 2C). Glucose content in whole bulbs was increased after treatment with GA_3_ at higher concentrations (2, 3 mL L^−1^). We also found differences in glucose content between upper and lower sprout portions as a consequence of one day of GA_3_ soaking. The highest glucose content (2.75-fold higher in comparison to control) was observed in lower sprout portions soaked in 1 mg L^−1^ GA_3_. At higher concentrations of GA_3_, there was a significant decrease in glucose content in lower sprout portions. There was no significant difference in glucose concentration in upper sprout portions among all GA_3_ treatments. In fact, the glucose content in upper sprout portions of control bulbs was more than two times higher than that in GA_3_-treated bulbs.

### 2.2. Effect of Chilling of Bulbs on Sugar Content in Different Parts of the Sprouted Bulbs

The content of soluble sugars, fructose, and glucose in bulbs was monitored immediately after the end of a 5 week chilling period (0 weeks at 24 °C) and after 1, 2, or 5 weeks of growth at 24 °C (Figure 3). The lowest total soluble sugar content was observed in bulbs collected immediately after cold treatment (7 °C). During further growth at 24 °C, the total soluble sugar content significantly increased during the first two weeks of culture and remained at a much higher level than in chilled bulbs.

The lowest glucose content was also observed in bulbs collected immediately after chilling treatment. Glucose content rapidly increased during the first week of growth at 24 °C, and kept rising during further cultivation. At the end of the fifth week, glucose content was five-fold higher than that measured immediately after chilling (Figure 3). In contrast to the total soluble sugar and glucose contents, the fructose content in bulbs did not change in the first two weeks after chilling. Compared to control, it significantly increased only 5 weeks after chilling.

We analyzed the sugar content of sprouted bulbs in lower and upper sprout portions of newly formed leaves and compared it to nonchilled bulbs (Figure 4). After five weeks of growth at 24 °C, the total soluble sugar content in the lower sprout portions of chilled bulbs was significantly higher than in control nonchilled bulbs. Contrarily, in upper sprout portions of chilled bulbs, the total soluble sugar content was significantly lower than that in the control. In addition, the glucose and fructose content in upper sprout portions of the chilled bulbs were significantly lower than that in the control.

Glucose was the dominant sugar in both nonchilled and chilled snake’s head fritillary bulbs. In control nonchilled sprouted bulbs, there was no difference in glucose concentration between lower and upper sprout portions. However, glucose content was 146% higher in lower sprout portions of chilled bulbs, compared to control nonchilled bulbs. We also recorded a significant difference in glucose concentration in lower and upper sprout portions when bulbs were subjected to low-temperature treatment (glucose lower:glucose upper = 3:1).

Histo-anatomical analysis did not reveal a distinctive difference between cold-pretreated and untreated bulbs. Microscopic observations showed that both chilled (Figure 5A) and nonchilled bulbs (Figure 5B,C) contained densely packed cells, of about 30 μM in diameter. Starch storage bodies, amyloplasts, containing numerous starch granules, were clearly observed in both nonchilled and chilled bulbs (Figure 5A).

## 3. Discussion

There is a lot of evidence that bulb treatment with GA_3_ solution affects bulb sprouting and further development [7,10,11,12]. Liu et al. [10] showed that soaking seed cloves in 1 mg mL^−1^ GA_3_ solution strongly induced lateral bud formation and clove number per bulb, and that gibberellins changed the shape and structure of bulbs in *Allium sativum*. Similarly, soaking *F. meleagris* bulbs in GA_3_ increased the average number of bulbs after cold pretreatment at chilling temperatures [1]. After soaking in GA_3_, *A. sativum* bulbs had significantly increased bulb weight and volume, and the plants had a higher incidence rate compared to control [10]. Similar changes, although to a lesser extent, were observed in *F. meleagris* bulbs in vitro. Soaking cloves in GA_3_ solution induced dormancy break and enhanced sprouting in garlic [7], which is in accordance with our results. In snake’s head fritillary, soaking bulbs in GA_3_ solution enhanced bulb development and growth, but soaked bulbs eventually underwent necrosis, whereas bulbs cultured on the medium with GA_3_ displayed no necrotic lesions [13]. Therefore, if not preceded by a chilling treatment, GA_3_ soaking treatment proved to be less satisfactory for the long-term propagation and breaking dormancy in vitro.

Starch and fructan were the major nonstructural carbohydrates detected in dormant storage organs in a number of ornamental geophytes, while soluble sugars (glucose, fructose, and sucrose) were present at much lower concentrations [11]. The total soluble sugar and glucose content in *F. meleagris* bulbs were higher in GA_3_-soaked bulbs than in control, water-soaked bulbs, indicating an important role of glucose in dormancy release and sprouting. A similar increase in soluble sugar content was observed in stems of GA_3_-treated *A. sativum* plants [10]. It is reasonable to suppose that the huge increase in soluble sugar content in GA_3_-treated bulbs marks the onset of sprouting and potentially the initiation of axillary meristem.

The main sugar in GA_3_-soaked *F. meleagris* bulbs was glucose, and its concentration varied during morphogenesis, dormancy, and sprouting. A sharp increase in glucose concentration after one week, in soaked bulbs, could be attributed to faster sugar hydrolysis when bulbs were soaked in GA_3_ solution because of the greater bulb surface being in contact with plant growth regulator compared to culture medium. Increased sugar concentration in soaked bulbs could be the result of mobilizing the accumulated reserve. Synthesis and accumulation of starch in *Tulipa* bulbs occurred rapidly in the middle and later swelling stages, in parallel with the decrease in sucrose content and endogenous GA level [5]. Exogenous addition of GA_3_, in our study, resulted in an increase in the total sugar and glucose content, leading to the onset of sprouting.

Generally, sucrose degradation provides energy for growth and for sucrose gradient, which will enable sucrose filling of the phloem and create the conditions for starch synthesis for new bulbs formation [12]. In bulbs soaked in water, there was no difference in sugar composition between the bulb and lower sprout portion. In these bulbs, there was no sugar gradient between different parts of the plant. A sugar gradient in different parts of the sprouted *F. meleagris* bulbs was also observed after cold treatment. Bulbs without GA_3_ treatment had no sugar gradient, their sprouting was inadequate, and the subsequent plant growth ceased at some point [1].

Histological analysis did not reveal a distinctive difference between chilled and nonchilled bulbs. In bulbs cultured at both temperatures, cells were characterized by an abundance of amyloplasts, containing starch granules. Scanning electron microscope observation of amyloplasts during the bulb development in *Hyacinthus orientalis* also revealed no differences between cold-treated and untreated cultures during the first two weeks of culture [14]. The number and the size of amyloplasts were shown to increase during bulb dormancy [8,15]. An abundance of amyloplasts was detected in nonsprouted bulbs even if they were treated with GA_3_ [13]. When bulbs started to sprout, the number of amyloplasts decreased even if they were not subjected to low temperature [13]. Starch degradation in hyacinth began gradually after 2 weeks and the number of the deposited starch granules was greatly reduced after 16 weeks of culture, suggesting that the starch had been utilized as an energy source for growth of the bulblets during their development [14].

Soluble carbohydrates, predominantly sucrose, have been considered as inducers of bulb formation [16]. In *F. meleagris* bulbs, glucose and fructose levels were lower in bulbs cultured at 7 °C, which supports the finding that sucrose was the main sugar in dormant bulbs. The total soluble sugar and glucose concentration in *F. meleagris* bulbs rapidly increased in the first week after chilling and remained relatively high over the course of the next four weeks. After five weeks of morphogenesis, significantly higher levels of all investigated sugars were detected in the lower sprout portion of chilled bulbs. Glucose accumulation in the lower sprout portion could be a result of starch degradation initiated at the end of low-temperature treatment and might reflect the inability of the plant to utilize a high amount of sugar at the very beginning of sprouting. Glucose content in the upper sprout portion of chilled bulbs was remarkably lower compared to the lower sprout portion, and further investigations will aim to explain why. In nonchilled bulbs, glucose content did not differ between the upper and lower sprout portions, corroborating a huge influence of chilling on fritillary physiology.

The logical next step for further experiments will be to examine sugar distribution over an extended period of time after chilling and track potential changes in sugar composition and localization. Future experiments should also include chilling treatment in combination with GA_3_ soaking to examine their effect on sugar composition, bulb sprouting, and their further development, in an attempt to further clarify the mechanisms underlying the bulb dormancy.

## 4. Material and Methods

### 4.1. Plant Material and Culture Conditions

Bulbs of *F. meleagris* L. multiplied on medium with Murashige and Skoog mineral solution [17] supplemented with 3% sucrose, 0.7% agar (MS medium), and 1.0 mg L^−1^ thidiazuron were used for this study. Culture medium was adjusted to 5.8 with 1 *n* NaOH before autoclaving. Cultures were grown at 24 ± 2 °C and a 16 h light/8 h dark photoperiod with an irradiance of 40 µmol m^−2^ s^−1^ in the culture room.

#### 4.1.1. Bulb Soaking with GA_3_

Two months after the last subculture, bulblets (~100 mg each) were collected and soaked in GA_3_ solution for 24 h, according to the procedure described by Petrić et al. [1]. GA_3_ was filter-sterilized by filtration through a 0.2 μM Dynagard membrane at the following final concentrations: 1, 2, and 3 mg L^−1^. Control bulbs were soaked in sterile distilled water for 24 h. After 24 h of GA_3_ treatment, bulbs were placed on MS medium without growth regulators and cultivated for one month (4 weeks) before sugar content analysis.

#### 4.1.2. Chilling of Bulbs

Isolated bulbs were grown on PGR-free MS medium in a growth chamber at 7 °C for five weeks. After this period of chilling, bulbs were grown for 0, 1, 2, and 5 five weeks on the same culture medium at room temperature (24 °C). Control bulbs were not exposed to cold treatment.

### 4.2. Morpho-Anatomical Analysis

Bulbs were fixed in FAA fixative (5 mL of 40% formalin, 5 mL of glacial acetic acid, and 90 mL of 70% ethanol [18]), at 4 °C for 24 h. Fixed material was washed, dehydrated in a graded ethanol series, and embedded in paraffin. Cross-sections (5−7 μM thick) were stained with hematoxylin or alcian blue, and photographed using a Leitz DMRB microscope (Leica, Wetzlar, Germany).

### 4.3. Sugar Analysis

Sugar content (total soluble sugars, fructose, and glucose) was measured in bulbs, lower and upper portions of sprouted bulbs after GA_3_ soaking (at all examined concentrations), and 4 weeks of growth at 24 °C. In addition, soluble sugar, fructose, and glucose contents in cold-pretreated bulbs were determined after 1, 2, and 5 weeks of growth at 24 °C. In addition, sugar content was measured in lower and upper sprout portions of cold-pretreated bulbs.

#### 4.3.1. Determination of Total Sugar Content

The total soluble sugar content was determined according to the modified Dreywood method [19]. Tissue samples (100 mg) were boiled for 3 h with 2.5 N HCl, and after cooling, Na_2_CO_3_ was added for neutralization. The material was also centrifuged (5000× *g*) for 2 min, and the supernatant was mixed with 0.2% solution of anthrone (Sigma Aldrich, St. Louis, MO, USA) in concentrated sulfuric acid (4 mL) and boiled for 10 min. After cooling, the total soluble sugar content was determined by measuring the absorbance at 620 nm (Thermo Scientific Multiskan FC, Waltham, MA, USA) using glucose as a standard.

#### 4.3.2. Fructose Determination

Fructose determination was done according to the modified Messineo and Musarra [20] protocol. Briefly, 2.5 mL of H_2_SO_4_ (75%) was added to 50 mg of plant material, and vortexed. After this step, 0.1 mL cysteine hydrochloride (2.5%) was added and vortexed again. The mixture was placed in a water bath at 45−50 °C during 10 min and a green color complex appeared. After cooling, 1 mL tryptophan hydrochloride solution was added to the mixture, and after one hour, the absorbance of the solution was measured at 518 nm. A standard solution of fructose was used for generating the standard curve.

#### 4.3.3. Glucose Determination

Glucose determination in samples was done by the modified method of Amaral et al. [21]. One mL of extraction mixture (0.25 mg ortho-dianisidine in 1 mL methanol, 0.1 M phosphate buffer, peroxidase, and glucose oxidase (Sigma Aldrich)) was added to 100 mg of plant material, and centrifuged. The sample was incubated for 40 min in a water bath at 35 °C. After incubation, reaction products had a pink color, the intensity of which was measured at 540 nm, after the reaction was stopped by the addition of 2 mL 6 N HCl. The intensity of the color of the solution was proportional to the glucose concentration in the sample.

### 4.4. Statistical Analysis of Data

The results of all experiments are presented as mean values ± standard errors of three independent experiments. The population, which was used in all experiments, was 30 bulbs per treatment. Three samples per GA_3_ concentration were used. Statistical analyses were performed using StatGraphics software version 4.2. Data were subjected to one-way analysis of variance (ANOVA) and significant differences between the means from each treatment were determined using Fisher’s least significant difference (LSD) test calculated at the confidence level of *p* ≤ 0.05. The graphical representation of the results was done using MS Excel.

## 5. Conclusions

In conclusion, soaking bulbs in 1 mg L^−1^ GA_3_ solution for 24 h and treatment with low temperatures caused significant change in the sugars status of snake’s head fritillary bulbs. The total soluble sugar and glucose content, predominantly in lower sprout portions, could be recognized as a signal for breaking dormancy of in vitro bulbs and pave the way for further investigations of the influence of gibberellins on dormancy. Studies concerning regeneration, sprouting, and dormancy breaking could very much improve propagation and development, as well as our understanding of the physiology of this process.

## Figures and Tables

**Figure 1 plants-09-01573-f001:**
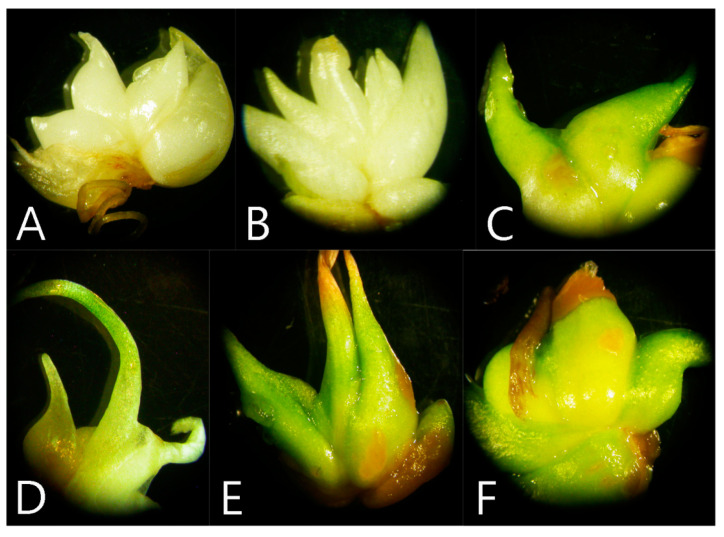
Effect of 24 h GA_3_ soaking treatment on bulb sprouting. (**A**) Control bulb soaked in sterile distilled water for 24 h after four weeks of culture, (**B**) bulb soaked in 1 mg L^−1^ GA_3_ after two weeks of growth, (**C**) bulb soaked in 2 mg L^−1^ GA_3_ after two weeks of growth, (**D**–**F**) sprouted bulbs after four weeks of culture, initially soaked in 1 mg L^−1^ GA_3_ (**D**), 2 mg L^−1^ GA_3_ (**E**), and 3 mg L^−1^ GA_3_ (**F**).

**Figure 2 plants-09-01573-f002:**
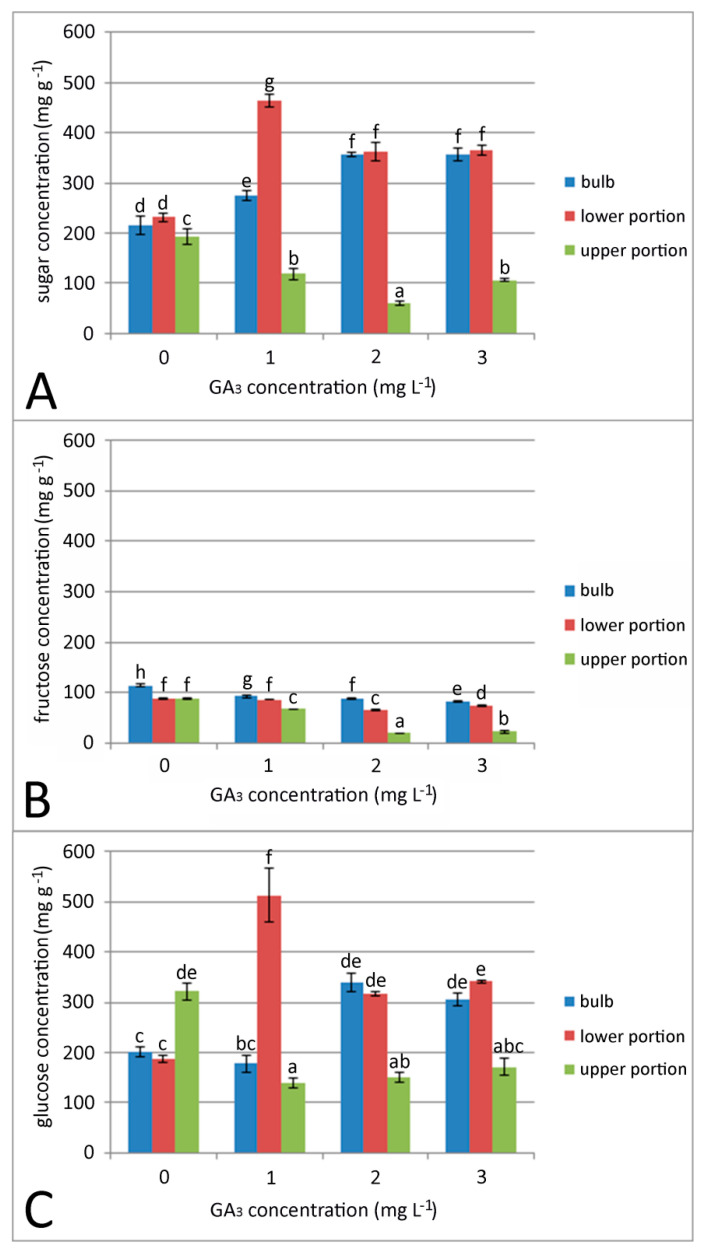
Effect of 24 h GA_3_ soaking treatment on total soluble sugar (**A**), fructose (**B**), and glucose (**C**) content in bulbs and upper and lower sprout portion. Control bulbs (0 mg L^–1^ GA_3_) were soaked in distilled water for 24 h. Means are expressed in mg/g ± SE. Data are means of independent measurements determined in triplicate. Different letters indicate significant differences between treatments at *p* < 0.05 (least significant difference (LSD) test).

**Figure 3 plants-09-01573-f003:**
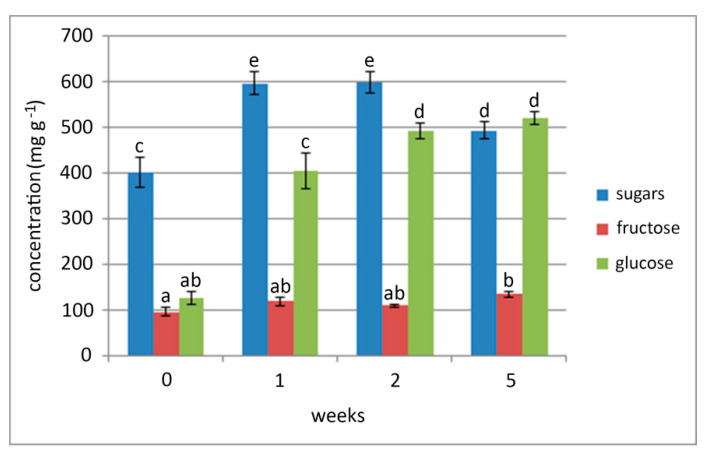
Total soluble sugar, fructose, and glucose content in chilled bulbs (7 °C, 5 weeks) after 0, 1, 2, and 5 weeks of growth at 24 °C. Chilled control bulbs were collected immediately after bulb chilling (0 weeks at 24 °C). Means are expressed in mg/g ± SE. Data are values of independent measurements determined in triplicate. Different letters indicate significant differences between treatments at *p* < 0.05 (LSD test).

**Figure 4 plants-09-01573-f004:**
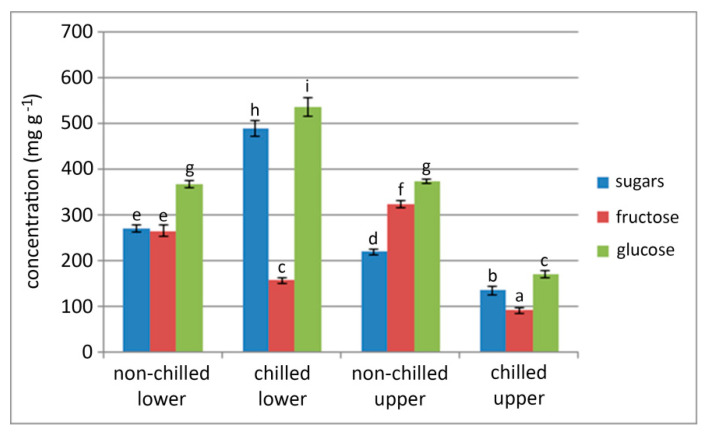
Total soluble sugar, fructose, and glucose content in lower and upper portions of sprouted bulbs obtained after chilling. Bulbs were chilled at 7 °C during 5 weeks and grown for 5 weeks at 24 °C, while control (nonchilled) bulbs were continually grown at 24 °C. Results are expressed in mg/g fresh weight ± SE. Data are means of independent measurements determined in triplicate. Different letters indicate significant differences between treatments at *p* < 0.05 (LSD test).

**Figure 5 plants-09-01573-f005:**
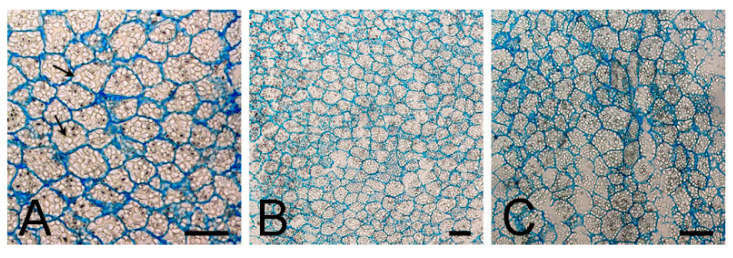
Histo-anatomical analysis of chilled and nonchilled bulbs grown on medium without plant growth regulators (PGRs). (**A**) Cross-section of the lower sprout portion of bulb cultured at 7 °C, (**B**,**C**) cross-sections of bulbs grown continually at 24 °C. Note the starch granules deposited in the amyloplasts (*arrows*). Scale bars = 100 μM.

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
