# Peer review of "Breaking the Dormancy of Snake’s Head Fritillary (Fritillaria meleagris L.) In Vitro Bulbs—Part 2: Effect of GA3 Soaking and Chilling on Sugar Status in Sprouted Bulbs"

_plants, 2020, doi:10.3390/plants9111573_

Round 1

Reviewer 1 Report

The manuscript “Breaking the dormancy of snake’s head fritillary (Fritillaria meleagris L.) in vitro bulbs – Part 2: Effect of GA3 soaking and chilling on sugar status in sprouted bulbs” by Marković et al describes the effects of different concentration of GA3 and low temperature treatment on the break of the dormancy in fritillary bulbs. The results agree with a similar study carried out in garlic (Lui et al. 2019 Sci. Hortic.).

The manuscript is interesting, but it has to be improved.

1) There are a lot of English mistakes. I suggest a revision by a native speaker or an English editing service. For example, in line 14 “combine” must be changed in “combined”; in line 16 “were analyzed” with “was analyzed”. In the same line “were determined” must be removed.

2) the authors investigated only the total soluble sugars, glucose and fructose content. I suggest extending the analysis to the other carbohydrates (such as sucrose, galactose, maltose, fructans, starch).

3) the authors must provide a possible mechanism of action that explains the effects of the gibberellins and cold treatment on the dormancy break in fritillary bulbs.

4) the introduction of some molecular analyses (i.e. RT-PCR on key genes involved in carbohydrate metabolism or allocation) could give more originality to the manuscript.

Author Response

Response to Reviewer 1 Comments

Thank you very much for reading the manuscript. Your comments and suggestions are very much appreciated. 

Reviewer’s comments and suggestions for Authors are underlined, and the appropriate response given below.

The manuscript “Breaking the dormancy of snake’s head fritillary (Fritillaria meleagris L.) in vitro bulbs – Part 2: Effect of GA3 soaking and chilling on sugar status in sprouted bulbs” by Marković et al describes the effects of different concentration of GA3 and low temperature treatment on the break of the dormancy in fritillary bulbs. The results agree with a similar study carried out in garlic (Lui et al. 2019 Sci. Hortic.).

The study of Liu et al (2019) and many others encouraged us to think about this kind of experiment in F. meleagris in an attempt to elucidate some aspects of bulb dormancy and sugar status. The results and design of the experiment were in accordance with our previous research (Petric et al 2013), but with some modifications in GA3 and GA inhibitors application. We submitted another manuscript, Part 1, which deals with sugar composition, bulb sprouting and histology of sprouted bulbs. The methods described in Part 1, as well as the findings presented in it, precede those in Part 2 and are valuable for its better understanding. Together these two manuscript allow for better understanding of breaking dormancy in vitro.

1) There are a lot of English mistakes. I suggest a revision by a native speaker or an English editing service. For example, in line 14 “combine” must be changed in “combined”; in line 16 “were analyzed” with “was analyzed”. In the same line “were determined” must be removed.

All changes were accepted. Also, English was corrected throughout the text.

2) the authors investigated only the total soluble sugars, glucose and fructose content. I suggest extending the analysis to the other carbohydrates (such as sucrose, galactose, maltose, fructans, starch).

We are grateful for your observation and suggestion, which is scientifically well supported. Despite our initial plans to investigate a number of carbohydrates, we decided to reduce the extent of experiment in order to reduce the required production time for plant material and to obtain better results with the material we have.

We investigated total sugar content and glucose and fructose separately. This experiment (together with experiment number 1 (Part 1 of this manuscript)) was very extensive and long lasting – approximately between five weeks and 6 months, if we take into account the whole morphogenesis process and regeneration of initial explants. This was explained in Material and Method section, but not in this way of timeline) for these three measurements (total sugar, glucose and fructose determination). Every sample weighed approximately 100 mg; multiplied with 30 (triplicates for each measurement), this would require a lot of plant in vitro material from the initial  in vitro stock. This, in turn, would require more time for further multiplication of plant material as well as for sugar determination. So, after careful consideration, our team decided to do these three measurements for this Article. We consider that these results explain the beginning  of sprouting and dormancy to a certain extent.

We certainly plan to continue this research and to include other sugars in the analysis as you suggest.

3) the authors must provide a possible mechanism of action that explains the effects of the gibberellins and cold treatment on the dormancy break in fritillary bulbs.

We did not attempt to deal with the mechanism of action of gibberellin because that is a complex mechanism that needs to be investigated at the molecular level. The title of our Article is therefore: Breaking the dormancy of snake’s head fritillary (Fritillaria meleagris L.) in vitro bulbs – Part 2: Effect of GA3 soaking and chilling on sugar status in sprouted bulbs and it is submitted for publication in the special issue Plant Tissue Culture. We investigated only sugar status during different time in bulbs, accompanied with histological analysis, in order to elucidate morphogenetic changes after different treatments and to improve in vitro propagation, We believe that we are not competent enough to infer about the potential mechanism without molecular investigation. Such an experiment will extend our work and is envisaged in the future. Your observation is very valuable and very logical from the scientific point of view, and remains a task to be performed in some future research. While designing the experiment, we relied mostly on papers by Liu et al. (2019), Petric et al. (2013), Rahman et al. (2006), who explained the effect of gibberellin without molecular mechanism. Also, in Part 1 of this paper (which, unfortunately, you did not get to read) we investigated sugar status after different cooling treatments and different time duration of temperatures, sprouting percentages and fresh weight after gibberellin and GA inhibitors treatment. The modes of administration differed in these two studies: in Part 1 experiments, GA3 and inhibitors were contained in nutrition medium, while in Part 2 experiments, the bulbs were soaked in GA3 solution. Taken together, Part 1 and Part 2 provide a bigger picture of the sprouting process. We wanted to show which was the main sugar in bulbs, how sugars were distributed during different times and treatments, how GA3 influenced sprouting capacity and sugar distribution in bulbs.

4) the introduction of some molecular analyses (i.e. RT-PCR on key genes involved in carbohydrate metabolism or allocation) could give more originality to the manuscript.

As we stated above, your suggestion is very logical and it would very much improve understanding of the mechanism of gibberellin action. Methods of molecular analysis are very attractive and very desirable from physiological point of view, as they allow for elucidation of molecular mechanisms of gibberellin action.

As said, we do consider some molecular analyses for further experiments, but such analyses require substantial resources that we were not able to acquire at this point. Here we focused on the effect of GA3 soaking on sugar status and distribution, morphogenesis and breaking dormancy for better understanding of F. meleagris multiplication in vitro. The aim of this work (Part 1 and Part 2 combined) was faster regeneration of explants, and optimization of methods of GA3 application. These two papers were intended for submission in the special issue, such as Plant tissue culture or any other dealing with in vitro morphogenesis and regeneration improvement.

Reviewer 2 Report

In the manuscript entitled “Breaking the dormancy of snake’s head fritillary (Fritillaria meleagris L.) in vitro bulbs – Part 2: Effect of GA3 soaking and chilling on sugar status in sprouted bulbs” the authors determined the sugar content on soaking bulbs in GA3 solution upon braking dormancy.

The work does not show novelty, the histological pictures and description is poor, and the English requires a major review.

Below, are only some examples of what needs to be improved.

Page 1, line 14 – “combined”

Page 1, line 15-16 – was analyzed. Eliminate “were determined”

Page 1, line 20 – during the first…

Page 1, line 21 – indicates the importance of low….

Page 3, lines 88-92. Too much information in just one sentence. It’s very confusing.

Figure 2 – Every single graph must contain all the legends, i.e., “GA3 concentration (mg/L)”

Page 5, line 120 – How long did the chilling period last? All these details need to be explained in the text.

Page 6, lines 158-160 – It is possible to state what the authors just wrote in this sentence based on the pictures presented. You ought to show higher magnification pictures in order to proper show the amyloplasts and the cell wall. The quality of the pictures is not the best one, showing dye deposits.

Page 6, line 169 – What does the authors mean with “quality of bulbs”? Quality in what terms? This is very subjective.

In the discussion section, the authors show a circular idea of the results. They have to read more and discuss the relation between GA synthetic pathway, carbohydrates reserves, sugar metabolization and how the pathways unfold. Association with dormancy break in bulbs and for example, in trees growing in boreal climates, should be addressed.

Author Response

Response to Reviewer 2 Comments

Thank you very much for reading the manuscript. Your comments and suggestions are very much appreciated, as they helped us greatly to improve the text. 

Reviewer’s comments and suggestions for Authors are underlined, and the appropriate response given below.

Comments and Suggestions for Authors

In the manuscript entitled “Breaking the dormancy of snake’s head fritillary (Fritillaria meleagris L.) in vitro bulbs – Part 2: Effect of GA3 soaking and chilling on sugar status in sprouted bulbs” the authors determined the sugar content on soaking bulbs in GA3 solution upon braking dormancy.

The work does not show novelty, the histological pictures and description is poor, and the English requires a major review.

According to your suggestion, we performed major review of the whole Article and made English corrections. We also submitted one more Article (Part 1, which will be connected with Part 2), and explained better morphology of plants in vitro under different treatments (GA3 and GA inhibitors, chilling temperatures) and their effects on sprouting capacity and potential weight impact. In Part 1 we submitted micrographs of processed bulbs after different treatments, while in Part 2 we only presented those of bulbs cultured without growth regulators at two different temperatures. No huge morphological changes were observed and we have therefore chosen only one Figure in this part of the article to depict the difference between treated and non treated bulbs. Contrary to control bulbs, bulbs cultured with gibberellin and under cold treatment were easier for processing hence their micrographs and descriptions were much better.

Below, are only some examples of what needs to be improved.

Page 1, line 14 – “combined”

Page 1, line 15-16 – was analyzed. Eliminate “were determined”

Page 1, line 20 – during the first…

Page 1, line 21 – indicates the importance of low….

All suggested corrections were introduced in text.

Page 3, lines 88-92. Too much information in just one sentence. It’s very confusing.

„Soluble sugar (Figure 2A), fructose (Figure 2B) and glucose (Figure 2C) contents were analyzed in bulbs and bottom and upper part of sprouted bulbs four weeks after soaking in GA3 solution (1, 2 and 3 mg L-1). Results were compared with control bulbs soaked in water for 24 h (Figure 1 A-C). We found significant changes in all sugars, as well as in different parts of sprouted bulbs effected by one day soaking in GA3.“

Figure 2 – Every single graph must contain all the legends, i.e., “GA3 concentration (mg/L)”

Figure 2 was corrected according to the reviewer’s suggestion.

Page 5, line 120 – How long did the chilling period last? All these details need to be explained in the text.

The chilling period lasted five weeks (as we described in Part 1, but seem to have omitted in Part 2). This information was added to the text (line 129 in the revised version).

Page 6, lines 158-160 – It is possible to state what the authors just wrote in this sentence based on the pictures presented. You ought to show higher magnification pictures in order to proper show the amyloplasts and the cell wall. The quality of the pictures is not the best one, showing dye deposits.

We have modified this figure according to the reviewer’s suggestion, replacing two lower-magnification micrographs with higher-magnification ones, including the scale bars. We believe that in all these micrographs abundant amyloplasts, filled with starch grains, are clearly visible. We have also slightly modified the text depicting this figure [line 174 in the revised version].  

Page 6, line 169 – What does the authors mean with “quality of bulbs”? Quality in what terms? This is very subjective.

By ’quality’ of the bulbs we meant their overall morphology, sprouting capacity, the absence of necrotic tissue and growth altogether. We obviously failed to find an adequate expression that would comprise all these traits, since the term ’quality’ appears to be very confusing. Upon your suggestion, we changed it into ’morphology’ (lines 183, 184 in the revised version).

In the discussion section, the authors show a circular idea of the results. They have to read more and discuss the relation between GA synthetic pathway, carbohydrates reserves, sugar metabolization and how the pathways unfold. Association with dormancy break in bulbs and for example, in trees growing in boreal climates, should be addressed.

We did not attempt to explain the mechanism of gibberellin action because that is a complex mechanism that needs to be investigated at the molecular level. The title of our article is therefore: Breaking the dormancy of snake’s head fritillary (Fritillaria meleagris L.) in vitro bulbs – Part 2: Effect of GA3 soaking and chilling on sugar status in sprouted bulbs. We investigated only sugar status during different time in bulbs with histological analysis in order to elucidate morphogenetic changes after different treatments. We believe that we are not competent enough to infer about potential mechanism without molecular investigation.

During experiment design, we mostly relied on papers by Liu et al. (2019), Petric et al. (2013), Rahman et al. (2006). They explained the effect of gibberellin without delving into molecular mechanism. Also, in Part 1 of this paper (which you did not get to read) we studied the sugar status after different treatments of cooling and different time duration of temperature treatments, sprouting percentages and fresh weight after gibberellin and GA inhibitors treatment. In Part 1 we investigated the influence of GA3 and inhibitors in nutrition medium and in this Part 2 we soaked bulbs in GA3 solution. Together, Part 1 and Part 2 showed a bigger picture of sprouting process. We wanted to show which was the main sugar in bulbs, how sugars were distributed over the course of time during different treatments, how GA3 influenced sprouting capacity and potential sugar distribution in bulbs.

Also, dormancy break in bulbs and trees in boreal climates are different and very hard to connect in discussion section. We attempted to associate other geophytes and their dormancy (with gibberellin influence) with our results on F. meleagris. Dormancy in trees is very complex and not yet elucidated, so we decided not to discuss such a complex process, but limited the discussion to our and related results.

Reviewer 3 Report

Revision of the manuscript ID: plants-964480

The manuscript  provides some interesting data on endogenous content of soluble sugars in Fritillaria meleagris L. bulbs influenced by exogenous treatments with GA3 and low temperature using in vitro system. In this system,  bulbs were treated by soaking in gibberellin solutions. The authors have shown the differences in the contents of particular types of sugars depending on the part of the bulb and in the process of forming new bulbs after low temperature treatment, with a simultaneous description of morphological differences in bulbs developing under different conditions. It is basically a simple work (no histological analysis of anatomical changes during daughter bulb formation). The experiments were quite well planned. However, there are some ambiguities between the description in the methodology and the description of the results.

The manuscript should be corrected by any proof-reading service.

There are the following mistakes and some comments to authors.

Abstract

- Temperature: 7oC  (no space, “7 oC” is incorrect, please correct throughout the text)

Introduction

Each statement should be supported by references, eg. “The fritillary bulbs contain many biological compounds with medicinal properties….”

Lines 27-28: it should be “In geophytes, the active growth of the aboveground part is short……, whereas the winter period or dry season…

Lines 31-33: This statement is not true. There are many studies on the regulation of the geophyte storage organs formation or the dormancy development and release.

Lines 37-38: …translocation of sugars was documented before… For what geophyte species?

Lines 40-45: this fragment adds nothing. One introductory sentence on the role of various phytohormones in the regulation of a storage organs formation and dormancy is enough.

Line 49: It shoul be:…in combination with various temperature treatments.

Lines 52-53: Add references.

Lines 53 and 57: the statement that GA3 application by soaking in its solutions was unexplored (Line 53) contradicts the statement that: fritillary bulbs were soaked in GA3 solution in another study [ref No. 9] (Line 57).

Line 62: Rather: “Formation of storage organs such as corms, tubers or bulbs begin with swelling of various organs, bud, leaf, shoot or root. (add references).

Line 64: It should be: “…with carbohydrate accumulation,…”

Please change “substance” which is used in several places into “compound”.

Results

The results should be described more concisely.

Lines 130-135: This part is incomprehensible.

Figure 2B is without numbers showing GA3 concentration on the X axis.

Figure 3. The description is incomprehensible. The methodology says (line 303-304) that control bulbs were not exposed to cold treatment. But in the description of this figure, it is: “control bulbs were collected immediately after bulb chilling”. As I understand, the sugar content measurements were done  immediately after the end of cooling (0) and then 1, 3 and 5 weeks later. Results for control (uncooled bulbs) have not been presented. 

Discussion

The discussion is too long and contains extensive repetitions of the results. There are quite a few references to Part I - there is no description of what Part I is. In my opinion Discussion should be shorten to the half. The citations are very sparse, e.g. lines 162-163: “There is a lot of evidences…” and only one reference [10]....

Statements (line169) that bulb quality treated with GA3 was the same as those soaked in water is  contradictory to the description of the bulbs in Results (lines 72-82).

Lines 257-260: In Results and Materials and Methods, there is no information on amyloplast observation.

265-268: This part is incomprehensible.

The manuscript can be reconsider for publication as short communication after major revision and significantly shortening the discussion.

Alternatively, the article may be the original publication when combined with Part I. It is not clear what is the Part I of the work. As I understand from the discussion, the Part I includes some results of gibberellin treatment by its addition to a medium. If the Part I  is another article sent to the same Journal and has not been already published yet, I suggest combining Part I and Part II (current article), since a large part of the discussion concerns Part I. This would greatly facilitate understanding the aim of the research, discussion of the results and drawing conclusions.

Author Response

Comments and Suggestions for Authors

Response to Reviewer 3 Comments

Thank you for your relevant and useful comments. We are particularly grateful for drawing our attention to the significant flaws in our manuscript, which we have corrected to the best of our knowledge.

Reviewer’s comments and suggestions for Authors are underlined, and the appropriate response given below.

Revision of the manuscript ID: plants-964480

The manuscript  provides some interesting data on endogenous content of soluble sugars in Fritillaria meleagris L. bulbs influenced by exogenous treatments with GA3 and low temperature using in vitro system. In this system,  bulbs were treated by soaking in gibberellin solutions. The authors have shown the differences in the contents of particular types of sugars depending on the part of the bulb and in the process of forming new bulbs after low temperature treatment, with a simultaneous description of morphological differences in bulbs developing under different conditions. It is basically a simple work (no histological analysis of anatomical changes during daughter bulb formation). The experiments were quite well planned. However, there are some ambiguities between the description in the methodology and the description of the results.

Thank you very much for this comment. We also submitted another article (Part 1 of this experiment), the results of which are in accordance with the results from Part 2. In Part 1, we analyzed histological changes in bulbs under different treatments (GA3 and GA inhibitors during time) and provided micrographs of sprouted bulbs and meristem formation, showing anatomical changes during daughter bulb formation. In this paper, we decided to show only control bulbs cultured without any treatment. Part 1 and Part 2 together represent one big experiment with related results. Reviewers got to read papers separately and that is a very hard task to do.

The manuscript should be corrected by any proof-reading service.

The manuscript has been corrected by a proof-reading service, according to your suggestion.

There are the following mistakes and some comments to authors.

Abstract

- Temperature: 7oC  (no space, “7 oC” is incorrect, please correct throughout the text)

Spaces preceding degree sign have been removed throughout the text.

Introduction

Each statement should be supported by references, eg. “The fritillary bulbs contain many biological compounds with medicinal properties….”

We have now introduced appropriate references supporting the statements, as suggested by the reviewer.

Lines 27-28: it should be “In geophytes, the active growth of the aboveground part is short……, whereas the winter period or dry season…

A suggested modification of the sentence in lines 27-28 has been made.

Lines 31-33: This statement is not true. There are many studies on the regulation of the geophyte storage organs formation or the dormancy development and release.

Thank you for drawing our attention to this statement. We are aware of the existence of many such studies about dormancy, especially knowing that we ourselves have cited some of them in this very manuscript. The corrected sentence is in lines 31-33 in the revised version.

Lines 37-38: …translocation of sugars was documented before… For what geophyte species?

We have listed some geophyte species in which the translocation of sugars has been documented, as requested by the reviewer [lines 38-39 in the revised version of the manuscript].

Lines 40-45: this fragment adds nothing. One introductory sentence on the role of various phytohormones in the regulation of a storage organs formation and dormancy is enough.

Unnecessary sentence has been removed from the text, according to the reviewer’s suggestion.

Line 49: It shoul be:…in combination with various temperature treatments.

Suggested modification has been made [line 50 in the revised version of the manuscript].

Lines 52-53: Add references.

Appropriate references were added, as requested.

Lines 53 and 57: the statement that GA3 application by soaking in its solutions was unexplored (Line 53) contradicts the statement that: fritillary bulbs were soaked in GA3 solution in another study [ref No. 9] (Line 57).

Ref No. 9 [which is No. 1 in the revised version of the manuscript] is our study from 2013, in which bulbs were soaked in GA3 solution but without cold pretreatment, and without the analysis of sugar content, morphology or anatomy. That is why we used the term "unexplored", which is apparently confusing, Therefore, we corrected this statement by adding "mostly" to mitigate the claim [line 54 in the revised version of the manuscript].

Line 62: Rather: “Formation of storage organs such as corms, tubers or bulbs begin with swelling of various organs, bud, leaf, shoot or root. (add references).

Suggested corrections have been made [lines 64-65 in the revised version of the manuscript].

Line 64: It should be: “…with carbohydrate accumulation,…”

Corrected [line 67 in the revised version].

Please change “substance” which is used in several places into “compound”.

The term ‘substance’ has been changed into ‘compound’ [lines 30, 71 in the revised version of the manuscript].

Results

The results should be described more concisely.

Lines 130-135: This part is incomprehensible.

This part was rewritten in a more comprehensive manner [lines 140-145 in revised version].

Figure 2B is without numbers showing GA3 concentration on the X axis.

Numbers showing GA3 concentration on the X axis in Figure 2B were added, as suggested.

Figure 3. The description is incomprehensible. The methodology says (line 303-304) that control bulbs were not exposed to cold treatment. But in the description of this figure, it is: “control bulbs were collected immediately after bulb chilling”. As I understand, the sugar content measurements were done  immediately after the end of cooling (0) and then 1, 3 and 5 weeks later. Results for control (uncooled bulbs) have not been presented. 

The description of Figure 3 has been modified in a more comprehensive manner.

In the experiment, we have used non-cooled bulbs (grown at 24°C only) and bulbs collected immediately after chilling (0 weeks at 24°C) for different comparisons. In material and method section, we described control as non-cooled bulbs (Figure 4 and some Figures in Part 1), but for Figure 3 comparison was made among cooled bulbs cultured for different time periods after cooling (0, 1, 2 and 5 weeks). We corrected this explanation and indicated these bulbs:

“Total soluble sugar, fructose and glucose content in chilled bulbs (7°C, 5 weeks) after 0, 1, 2 and 5 weeks of growth at 24°C. Chilled control bulbs (0 weeks) were collected immediately after bulb chilling. Means are expressed in mg/g ± SE. Data are values of independent measurements determined in triplicate. Different letters indicate significant differences between treatments at P<0.05 (LSD test).“ [lines 135-139 in the revised version of the manuscript].

Control bulbs in Figure 4 [lines 157-161 in the revised version of the manuscript] are in accordance with the explanation in Material and method section  [lines 320-323 in the revised version of the manuscript].

Discussion

The discussion is too long and contains extensive repetitions of the results. There are quite a few references to Part I - there is no description of what Part I is. In my opinion Discussion should be shorten to the half. The citations are very sparse, e.g. lines 162-163: “There is a lot of evidences…” and only one reference [10]....

References were added in the above mentioned statement. Lines 176-178 in revised version.

Statements (line169) that bulb quality treated with GA3 was the same as those soaked in water is  contradictory to the description of the bulbs in Results (lines 72-82).

By ’quality’ of the bulbs we meant their overall morphology, sprouting capacity, the absence of necrotic tissue and growth altogether. We obviously failed to find an adequate expression that would comprise all these traits, since the term ’quality’ appears to be very confusing. Upon the suggestion of the Reviewer 2, we changed it into ’morphology’ (lines 183, 184 in the revised version). Differences refer to the color, sprouting capacity and the potential presence of the necrotic tissue, but morphology was the same.

Lines 257-260: In Results and Materials and Methods, there is no information on amyloplast observation.

Results, lines 172-173 in the revised version of the manuscript]: “ In bulbs cultured at both temperatures, cells were characterized by an abundance of amyloplasts (Figure 5).“

Material and Methods [lines 325-328 in the revised version of the manuscript]: we described histological analysis which was the same as in Part 1, where we analyzed amyloplasts after different treatments and culture conditions (in Part 2, we only presented bulbs cultured without PGRs).

265-268: This part is incomprehensible.

This part was rewritten in a more comprehensive manner [lines 275-279 in revised version].

The manuscript can be reconsider for publication as short communication after major revision and significantly shortening the discussion.

Alternatively, the article may be the original publication when combined with Part I. It is not clear what is the Part I of the work. As I understand from the discussion, the Part I includes some results of gibberellin treatment by its addition to a medium. If the Part I  is another article sent to the same Journal and has not been already published yet, I suggest combining Part I and Part II (current article), since a large part of the discussion concerns Part I. This would greatly facilitate understanding the aim of the research, discussion of the results and drawing conclusions.

Thank you again for reading the Article and for very useful suggestions. We submitted both manuscripts (Part 1 and Part 2) together, but reviewers got them separately. Part 1 is easier to read separately than Part 2, which is a logical next step of experiments. We understand how difficult task you had to read Part 2 without the results from Part 1. We intend to publish both manuscripts together in Plant Tissue Culture issue, which should allow better understanding of Part 2. Our goal in Part 1 was to explain and compare GA3, ancyimidol and paclobutrazol application in medium, and the effect of such administration of these PGRs on sugar status, chilling influence and morpho-anatomy. Part 2  is the second part of the experiment, where GA3 was administered only in soaking solution. We wanted to compare its application and its influence on sugars. The aim of both manuscripts together was the improvement of regeneration of F. meleagris in vitro. Because of that we relied on discussion from Part 1 to show differences and improvements. You are right when you say that Part 1 would greatly facilitate understanding the aim of research, discussion and conclusions of the Part 2. We hope that both manuscripts will be published together exactly because of that.

Round 2

Reviewer 1 Report

The authors have improved the munuscript.

Author Response

Thank you for helping us to correct and improve our manuscript.

Reviewer 2 Report

In the manuscript entitled “Breaking the dormancy of snake’s head fritillary (Fritillaria meleagris L.) in vitro bulbs – Part 2: Effect of GA3 soaking and chilling on sugar status in sprouted bulbs” the authors determined the sugar content on soaking bulbs in GA3 solution upon braking dormancy.

Upon revision by the authors, the English has improved even though, there are still grammar incorrections that need to be corrected such as in lines: 27-28, 93, 96, 161, 254, etc.

In the abstract, the authors wrote: …” its accumulation was predominantly detected in the bottom part of the leaves”. I reckon they intendent to say bulb and not leaves. Still on this note, the more I read the expression “bottom part” throughout the text, the worse it sounds. The only thing that comes to my mind are body parts. It’s not scientific language. I suggest something like; “lower half portion of the bulb” or simply “lower portion”.

Line 100 – write “greener”. The same should be corrected in the rest of the text.

Lines 168 and 232. Keep all text in the past tense form.

The authors simply did not address the problem of the histological part of the manuscript. The replaced one picture by another with the same bad quality. The description of the pictures did not suffer any improvement and there is no indication on the figures that guides the reader to what the authors are talking about. I know what an amyloplast looks like but most of the readers do not and the authors should not assume that is common knowledge. I cannot accept the manuscript with this section like this.

The discussion not only changed but has worsened. I don’t understand what Part 1/I means. The same was brought to my attention in the author’s reply. This manuscript, just like any scientific publication, has to stand by itself. This is not a romance where the reader can only understand the second chapter after reading the first one. I never came across with something like this. What should we do when one reads (Part 1)? I understand you are trying to get as many publications as possible from the same work but there is a price at the end of the road.

Author Response

Response to Reviewer 2 Comments

Comments and Suggestions for Authors

In the manuscript entitled “Breaking the dormancy of snake’s head fritillary (Fritillaria meleagris L.) in vitro bulbs – Part 2: Effect of GA3 soaking and chilling on sugar status in sprouted bulbs” the authors determined the sugar content on soaking bulbs in GA3 solution upon braking dormancy.

Upon revision by the authors, the English has improved even though, there are still grammar incorrections that need to be corrected such as in lines: 27-28, 93, 96, 161, 254, etc.

We have further improved English and corrected grammar errors throughout the manuscript.

In the abstract, the authors wrote: …” its accumulation was predominantly detected in the bottom part of the leaves”. I reckon they intendent to say bulb and not leaves. Still on this note, the more I read the expression “bottom part” throughout the text, the worse it sounds. The only thing that comes to my mind are body parts. It’s not scientific language. I suggest something like; “lower half portion of the bulb” or simply “lower portion”.

The term ‘bottom/upper part’ was replaced by “lower/upper sprout portion” throughout the text, and also in the figures (Fig. 2 and Fig. 4), according to the reviewer’s suggestion.

Line 100 – write “greener”. The same should be corrected in the rest of the text.

We have incorporated the suggested corrections.

Lines 168 and 232. Keep all text in the past tense form.

Corrected.

The authors simply did not address the problem of the histological part of the manuscript. The replaced one picture by another with the same bad quality. The description of the pictures did not suffer any improvement and there is no indication on the figures that guides the reader to what the authors are talking about. I know what an amyloplast looks like but most of the readers do not and the authors should not assume that is common knowledge. I cannot accept the manuscript with this section like this.

The section regarding histology has been rewritten and we hope that the description has been improved. Amyloplasts are now clearly denoted in the micrographs, and in the legend to the figure.

The discussion not only changed but has worsened. I don’t understand what Part 1/I means. The same was brought to my attention in the author’s reply. This manuscript, just like any scientific publication, has to stand by itself. This is not a romance where the reader can only understand the second chapter after reading the first one. I never came across with something like this. What should we do when one reads (Part 1)? I understand you are trying to get as many publications as possible from the same work but there is a price at the end of the road.

We have rewritten the entire section Discussion, avoiding any references to the Part 1 of the manuscript.

Reviewer 3 Report

The authors mainly corrected minor mistakes. My greatest objections claim concern the discussion. I still find it too long and speculative. There are many repetitions of results as well as the repeated and contradictory statements.

Detailed comments:

Minor mistakes:

  • Snake’s head fritillary: English plant names should be written lowercase: “snake’s head fritillary”.
  • if you mean a naturally occurring process, use in vivo,g. morphogenesis and not "ex vitro", e.g. line 31 and some other places.
  • Please correct the figure number (lines 98-102); it should be figure 1 and not figure 4.
  • If you mean exogenously applied compounds, please use “plant growth regulators” (PGRs) (e.g. line 70); “phytohormones” are rather used for endogenous compounds.

Lines 185-190: suggestion to change into: “Glucose content significantly increased in bulbs after chilling period. It was three-fold higher and four-fold higher when measured one week and five weeks after chilling, respectively, in comparison to the glucose content examined immediately after low temperature treatment. In contrast to total soluble sugar and glucose contents, the fructose content in bulbs did not change during the first two weeks after chilling, and significantly increased five weeks after cooling.” Please check the glucose content increase (“two-fold and three-fold” are incorrect, these were considerably higher)

Line 247: “…as studies of Liu et al. [7, 10, 11, 12]” is improper, it should be: “as studies of other authors [7, 10, 11, 12]...” 

Discussion

Lines 249-273: “….In addition, gibberellins changed the shape and structure of bulbs. Our results also showed differences between bulbs soaked in water and those soaked in GA3 solution (regardless of the concentration). The formation of new bulblets on bulbs which were in GA3  solution were observed after four weeks. New bulblets were thinner and longer than bulbs soaked  in water. In our study, the morphology (shape) of bulbs soaked with GA3 was the same as the  morphology of bulbs soaked in water.???? This contrasted the findings for A. sativum bulbs, which were soaking in GA3, but this kind of regeneration has not been noticed in F. melegaris meleagris bulbs in vitro.???”  The last statements are contrary to those are stated above.

Lines 278-284: Additionally, soaking bulbs in GA3 solution proved to be a better choice:  for bulb development and growth of Ssnake’s head fritillary bulbs than supplementing solid growth medium with gibberellins (results from Part I)….

 …but soaked bulbs ceased growing at some point and underwent necrosis, while bulbs cultured on medium with GA3 displayed no necrotic lesions (Part 1). Therefore, culturing bulbs of F. meleagris on medium with GA3 is a better option for long-term propagation and breaking dormancy in vitro than soaking in GA3 solution.???”  This statement is contrary to that is stated above

Line 285: “Soaking in GA3 was shown to increase an average number of bulbs after cold pretreatment ???…”. Results on bulb numbers were not presented.

Lines 305-311: This fragment is completely incomprehensible.

Lines 359- 364: “In theory, the main carbohydrate in dormant storage organ is starch with a small percentage of soluble sugars, glucose and fructose [8]. Starch was hydrolyzed during growth into soluble sugars. In practice, many geophytes had high percentage of soluble sugars, instead of starch, in their storage organs when bulbs start to sprout, which were used to provide energy (required) for sprouting [11] as in fritillary bulbs.” This statement is incorrect. In all geophytes, where starch is the reserve form of carbohydrates, after dormancy release, starch is slowly converted to soluble sugars, with sucrose being a transport form and glucose and fructose are converted into other compounds. In Fig 5, amyloplasts with starch grains are present in fritillary in both chilled and uncooled bulbs. The intensity of this process depends on genotype and conditions.

Lines 383-392: It is not always known what the individual sentences refer to: their own results, the results of other authors or Part I. A similar remark applies to several other fragments.

The discussion should be considerably shortened. The text should be stylistically corrected. Repetitions of results and statements should be eliminated. Please also correct any incorrect statements and conclusions. There are too few references, only 19.

The manuscript still requires major revision of the discussion.

Author Response

Response to Reviewer 3 Comments

Comments and Suggestions for Authors

The authors mainly corrected minor mistakes. My greatest objections claim concern the discussion. I still find it too long and speculative. There are many repetitions of results as well as the repeated and contradictory statements.

We have further improved the manuscript, especially the section Discussion, which was rewritten completely, and is now substantially shorter, less speculative and clearer. The repetitions of the results and the contrary statements have been removed.

Detailed comments:

Minor mistakes:

Snake’s head fritillary: English plant names should be written lowercase: “snake’s head fritillary”.

Plant name has been corrected throughout the text.

if you mean a naturally occurring process, use in vivo,g. morphogenesis and not "ex vitro", e.g. line 31 and some other places.

The term ‘ex vitro’ has been replaced by ‘in vivo’.

Please correct the figure number (lines 98-102); it should be figure 1 and not figure 4.

The figure number has been corrected [lines 83-87 in the revised version].

If you mean exogenously applied compounds, please use “plant growth regulators” (PGRs) (e.g. line 70); “phytohormones” are rather used for endogenous compounds.

The term ‘phytohormones’ has been replaced by ‘plant growth regulators’, to underline their exogenous application [line 53 in the revised version].

Lines 185-190: suggestion to change into: “Glucose content significantly increased in bulbs after chilling period. It was three-fold higher and four-fold higher when measured one week and five weeks after chilling, respectively, in comparison to the glucose content examined immediately after low temperature treatment. In contrast to total soluble sugar and glucose contents, the fructose content in bulbs did not change during the first two weeks after chilling, and significantly increased five weeks after cooling.” Please check the glucose content increase (“two-fold and three-fold” are incorrect, these were considerably higher)

The suggested correction has been incorporated in the text [lines 146-150 in the revised version].

Line 247: “…as studies of Liu et al. [7, 10, 11, 12]” is improper, it should be: “as studies of other authors [7, 10, 11, 12]...” 

This statement was corrected, as suggested by the reviewer [lines 193-194 in the revised version].

Discussion

Lines 249-273: “….In addition, gibberellins changed the shape and structure of bulbs. Our results also showed differences between bulbs soaked in water and those soaked in GA3 solution (regardless of the concentration). The formation of new bulblets on bulbs which were in GA3  solution were observed after four weeks. New bulblets were thinner and longer than bulbs soaked  in water. In our study, the morphology (shape) of bulbs soaked with GA3 was the same as the  morphology of bulbs soaked in water.???? This contrasted the findings for A. sativum bulbs, which were soaking in GA3, but this kind of regeneration has not been noticed in F. melegaris meleagris bulbs in vitro.???”  The last statements are contrary to those are stated above.

The section Discussion was rewritten, and this entire paragraph was removed.

Lines 278-284: Additionally, soaking bulbs in GA3 solution proved to be a better choice:  for bulb development and growth of Ssnake’s head fritillary bulbs than supplementing solid growth medium with gibberellins (results from Part I)….

 …but soaked bulbs ceased growing at some point and underwent necrosis, while bulbs cultured on medium with GA3 displayed no necrotic lesions (Part 1). Therefore, culturing bulbs of F. meleagris on medium with GA3 is a better option for long-term propagation and breaking dormancy in vitro than soaking in GA3 solution.???”  This statement is contrary to that is stated above

The difference between modes of GA3 application has now been explained more clearly, without contradictory statements [lines 201-206 in the revised version].

Line 285: “Soaking in GA3 was shown to increase an average number of bulbs after cold pretreatment ???…”. Results on bulb numbers were not presented.

This finding is the result of our earlier study, which has now been clearly indicated by the reference number [line 197 in the revised version].

Lines 305-311: This fragment is completely incomprehensible.

The entire fragment has been removed.

Lines 359- 364: “In theory, the main carbohydrate in dormant storage organ is starch with a small percentage of soluble sugars, glucose and fructose [8]. Starch was hydrolyzed during growth into soluble sugars. In practice, many geophytes had high percentage of soluble sugars, instead of starch, in their storage organs when bulbs start to sprout, which were used to provide energy (required) for sprouting [11] as in fritillary bulbs.” This statement is incorrect. In all geophytes, where starch is the reserve form of carbohydrates, after dormancy release, starch is slowly converted to soluble sugars, with sucrose being a transport form and glucose and fructose are converted into other compounds. In Fig 5, amyloplasts with starch grains are present in fritillary in both chilled and uncooled bulbs. The intensity of this process depends on genotype and conditions.

Incorrect statement has been removed, and the entire section was rewritten [lines 207-230 in the revised version].

Lines 383-392: It is not always known what the individual sentences refer to: their own results, the results of other authors or Part I. A similar remark applies to several other fragments.

In this revised version of the manuscript, we have particularly paid attention to this problem. We avoided referencing to Part 1 for the most part; where there was necessary to compare modes of gibberellin application, we clearly denoted earlier results by citing the corresponding paper (Reference No.13 in the revised version).

The discussion should be considerably shortened. The text should be stylistically corrected. Repetitions of results and statements should be eliminated. Please also correct any incorrect statements and conclusions. There are too few references, only 19.

The manuscript still requires major revision of the discussion.

The section Discussion has been considerably shortened and corrected. Any repetitive or confusing statements, as well as those that were overly speculative have been removed. Two additional references have been introduced.

Round 3

Reviewer 2 Report

Review

In the manuscript entitled “Breaking the dormancy of snake’s head fritillary (Fritillaria meleagris L.) in vitro bulbs – Part 2: Effect of GA3 soaking and chilling on sugar status in sprouted bulbs” the authors determined the sugar content on soaking bulbs in GA3 solution upon braking dormancy.

Upon revision by the authors, the English has improved substantially. I have detected two small incorrections: line 35 – it should read…”source of energy.” Please, delete “the”.

Line 147 – this sentence needs to be rewritten since it is quite difficult to understand.

The discussion has also been improved and now reflects the rest of the manuscript.

Author Response

Response to Reviewer 2 Comments

Comments and Suggestions for Authors

Upon revision by the authors, the English has improved substantially. I have detected two small incorrections: line 35 – it should read…”source of energy.” Please, delete “the”.

This part of the sentence was corrected and it now reads “source of energy” (line 39 in the revised version).

Line 147 – this sentence needs to be rewritten since it is quite difficult to understand.

The sentence was modified for clarity (line 148- in the revised version).

The discussion has also been improved and now reflects the rest of the manuscript.

We are grateful for the valuable comments and suggestions that helped us improve this particular section.

Reviewer 3 Report

This revised version is much better. There are a few minor mistakes:

Line 29: it should be “…underground storage organ, e.g. bulb in fritillary.”

Line 40: it should be: “….are unknown in fritillary”.

  Line 102: should be: “…new bulblets were developed in control (Figure 1A). The regenerated bulblets…”

Line 363: Please explain the abbreviation “SEM”

I suggest to uniform the units: mg L-1 is used in text, but mg/L on graphs.

The manuscript can be published after correcting these minor mistakes.

Author Response

Response to Reviewer 3 Comments

Comments and Suggestions for Authors

Line 29: it should be “…underground storage organ, e.g. bulb in fritillary.”

The sentence was modified according to the reviewer’s suggestion (line 31 in the revised version).

Line 40: it should be: “….are unknown in fritillary”.

The sentence was modified according to the reviewer’s suggestion (lines 46-47 in the revised version).

Line 102: should be: “…new bulblets were developed in control (Figure 1A). The regenerated bulblets…”

The sentence was modified according to the reviewer’s suggestion (line 80 in the revised version).

Line 363: Please explain the abbreviation “SEM”

The abbreviation “SEM” was replaced by “Scanning electron microscope” (line 236 in the revised version).

I suggest to uniform the units: mg L-1 is used in text, but mg/L on graphs.

The units on the chart axes (Figures 2, 3 and 4) were modified to achieve the uniformity of units used on graphs and in the text.

The manuscript can be published after correcting these minor mistakes.
